# Non-Local Graph Neural Networks

## Abstract

Modern graph neural networks (GNNs) learn node embeddings through multi-layer local aggregation and achieve great success in applications on assortative graphs. However, tasks on disassortative graphs usually require non-local aggregation. In addition, we find that local aggregation is even harmful for some disassortative graphs. In this work, we propose a simple yet effective non-local aggregation framework with an efficient attention-guided sorting for GNNs. Based on it, we develop various non-local GNNs. We perform thorough experiments to analyze disassortative graph datasets and evaluate our non-local GNNs. Experimental results demonstrate that our non-local GNNs significantly outperform previous state-of-the-art methods on six benchmark datasets of disassortative graphs, in terms of both model performance and efficiency.

## 1 Introduction

Graph neural networks (GNNs) process graphs and map each node to an embedding vector (Zhang et al., 2018b; Wu et al., 2019). These node embeddings can be directly used for node-level applications, such as node classification (Kipf & Welling, 2017) and link prediction (Schütt et al., 2017). In addition, they can be used to learn the graph representation vector with graph pooling (Ying et al., 2018; Zhang et al., 2018a; Lee et al., 2019; Yuan & Ji, 2020), in order to fit graph-level tasks (Yanardag & Vishwanathan, 2015). Many variants of GNNs have been proposed, such as ChebNets (Defferrard et al., 2016), GCNs (Kipf & Welling, 2017), GraphSAGE (Hamilton et al., 2017), GATs (Veličković et al., 2018), LGCN (Gao et al., 2018) and GINs (Xu et al., 2019). Their advantages have been shown on various graph datasets and tasks (Errica et al., 2020). However, these GNNs share a multilayer local aggregation framework, which is similar to convolutional neural networks (CNNs) (LeCun et al., 1998) on grid-like data such as images and texts.

In recent years, the importance of non-local aggregation has been demonstrated in many applications in the field of computer vision (Wang et al., 2018; 2020) and natural language processing (Vaswani et al., 2017). In particular, the attention mechanism has been widely explored to achieve non-local aggregation and capture long-range dependencies from distant locations. Basically, the attention mechanism measures the similarity between every pair of locations and enables information to be communicated among distant but similar locations. In terms of graphs, non-local aggregation is also crucial for disassortative graphs, while previous studies of GNNs focus on assortative graph datasets (Section 2.2). In addition, we find that local aggregation is even harmful for some disassortative graphs (Section 4.3). The recently proposed Geom-GCN (Pei et al., 2020) explores to capture long-range dependencies in disassortative graphs. It contains an attention-like step that computes the Euclidean distance between every pair of nodes. However, this step is computationally prohibitive for large-scale graphs, as the computational complexity is quadratic in the number of nodes. In addition, Geom-GCN employs pre-trained node embeddings (Tenenbaum et al., 2000; Nickel & Kiela, 2017; Ribeiro et al., 2017) that are not task-specific, limiting the effectiveness and flexibility.

In this work, we propose a simple yet effective non-local aggregation framework for GNNs. At the heart of the framework lies an efficient attention-guided sorting, which enables non-local aggregation through classic local aggregation operators in general deep learning. The proposed framework can be flexibly used to augment common GNNs with low computational costs. Based on the framework, we build various efficient non-local GNNs. In addition, we perform detailed analysis on existing disassortative graph datasets, and apply different non-local GNNs accordingly. Experimental results show that our non-local GNNs significantly outperform previous state-of-the-art methods on node classification tasks on six benchmark datasets of disassortative graphs.

## 2 BACKGROUND AND RELATED WORK

### 2.1 GRAPH NEURAL NETWORKS

We focus on learning the embedding vector for each node through graph neural networks (GNNs). Most existing GNNs are inspired by convolutional neural networks (CNNs) (LeCun et al., 1998) and follow a local aggregation framework. In general, each layer of GNNs scans every node in the graph and aggregates local information from directly connected nodes, *i.e.*, the 1-hop neighbors.

Specifically, a common layer of GNNs performs a two-step processing similar to the depthwise separable convolution (Chollet, 2017): spatial aggregation and feature transformation. The first step updates each node embedding using embedding vectors of spatially neighboring nodes. For example, GCNs (Kipf & Welling, 2017) and GATs (Veličković et al., 2018) compute a weighted sum of node embeddings within the 1-hop neighborhood, where weights come from the degree of nodes and the interaction between nodes, respectively. GraphSAGE (Hamilton et al., 2017) applies the max pooling, while GINs (Xu et al., 2019) simply sums the node embeddings. The feature transformation step is similar to the $1 \times 1$ convolution, where each node embedding vector is mapped into a new feature space through a shared linear transformation (Kipf & Welling, 2017; Hamilton et al., 2017; Veličković et al., 2018) or multilayer perceptron (MLP) (Xu et al., 2019). Different from these studies, LGCN (Gao et al., 2018) explores to directly apply the regular convolution through top-$k$ ranking.

Nevertheless, each layer of these GNNs only aggregates local information within the 1-hop neighborhood. While stacking multiple layers can theoretically enable communication between nodes across the multi-hop neighborhood, the aggregation is essentially local. In addition, deep GNNs usually suffer from the over-smoothing problem (Xu et al., 2018; Li et al., 2018; Chen et al., 2020).

### 2.2 ASSORTATIVE AND DISASSORTATIVE GRAPHS

There are many kinds of graphs in the literature, such as citation networks (Kipf & Welling, 2017), community networks (Chen et al., 2020), co-occurrence networks (Tang et al., 2009), and webpage linking networks (Rozemberczki et al., 2019). We focus on graph datasets corresponding to the node classification tasks. In particular, we categorize graph datasets into assortative and disassortative ones (Newman, 2002; Ribeiro et al., 2017) according to the node homophily in terms of labels, *i.e.*, how likely nodes with the same label are near each other in the graph.

Assortative graphs refer to those with a high node homophily. Common assortative graph datasets are citation networks and community networks. On the other hand, graphs in disassortative graph datasets contain more nodes that have the same label but are distant from each other. Example disassortative graph datasets are co-occurrence networks and webpage linking networks.

As introduced above, most existing GNNs perform local aggregation only and achieve good performance on assortative graphs (Kipf & Welling, 2017; Hamilton et al., 2017; Veličković et al., 2018; Gao et al., 2018). However, they may fail on disassortative graphs, where informative nodes in the same class tend to be out of the local multi-hop neighborhood and non-local aggregation is needed. Thus, in this work, we explore the non-local GNNs.

### 2.3 ATTENTION MECHANISM

The attention mechanism (Vaswani et al., 2017) has been widely used in GNNs (Veličković et al., 2018; Gao & Ji, 2019; Knyazev et al., 2019) as well as other deep learning models (Yang et al., 2016; Wang et al., 2018; 2020). A typical attention mechanism takes three groups of vectors as inputs, namely the query vector $q$, key vectors $(k_1, k_2, \ldots, k_n)$, value vectors $(v_1, v_2, \ldots, v_n)$. Note that key and value vectors have a one-to-one correspondence and can be the same sometimes. The attention mechanism computes the output vector $o$ as

$$a_i = \text{ATTEND}(q, k_i) \in \mathbb{R}, \ i = 1, 2, \ldots, n; \quad o = \sum_i a_i v_i, \tag{1}$$

where the $\text{ATTEND}(\cdot)$ function could be any function that outputs a scalar attention score $a_i$ from the interaction between $q$ and $k_i$, such as dot product (Gao & Ji, 2019) or even a neural net-

work (Veličković et al., 2018). The definition of the three groups of input vectors depends on the models and applications.

Notably, existing GNNs usually use the attention mechanism for local aggregation (Veličković et al., 2018; Gao & Ji, 2019). Specifically, when aggregating information for node $v$, the query vector is the embedding vector of $v$ while the key and value vectors come from node embeddings of $v$'s directly connected nodes. And the process is iterated for each $v \in V$. It is worth noting that the attention mechanism can be easily extended for non-local aggregation (Wang et al., 2018; 2020), by letting the key and value vectors correspond to all the nodes in the graph when aggregating information for each node. However, it is computationally prohibitive given large-scale graphs, as iterating it for each node in a graph of $n$ nodes requires $O(n^2)$ time. In this work, we propose a novel non-local aggregation method that only requires $O(n \log n)$ time.

## 3 THE PROPOSED METHOD

### 3.1 NON-LOCAL AGGREGATION WITH ATTENTION-GUIDED SORTING

We consider a graph $\mathcal{G} = (V, E)$, where $V$ is the set of nodes and $E$ is the set of edges. Each edge $e \in E$ connects two nodes so that $E \subseteq V \times V$. Each node $v \in V$ has a node feature vector $x_v \in \mathbb{R}^d$. The $k$-hop neighborhood of $v$ refers to the set of nodes $\mathcal{N}_k(v)$ that can reach $v$ within $k$ edges. For example, the set of $v$'s directly connected nodes is its 1-hop neighborhood $\mathcal{N}_1(v)$.

Our proposed non-local aggregation framework is composed of three steps, namely local embedding, attention-guided sorting, and non-local aggregation. In the following, we describe them one by one.

**Local Embedding:** Our proposed framework is built upon a local embedding step that extracts local node embeddings from the node feature vectors. The local embedding step can be as simple as

$$z_v = \text{MLP}(x_v) \in \mathbb{R}^f, \ \forall v \in V. \tag{2}$$

The $\text{MLP}(\cdot)$ function is a multilayer perceptron (MLP), and $f$ is the dimension of the local node embedding $z_v$. Note that the $\text{MLP}(\cdot)$ function is shared across all the nodes in the graph. Applying MLP only takes the node itself into consideration without aggregating information from the neighborhood. This property is very important on some disassortative graphs, as shown in Section 4.3.

On the other hand, graph neural networks (GNNs) can be used as the local embedding step as well, so that our proposed framework can be easily employed to augment existing GNNs. As introduced in Section 2.1, modern GNNs perform multilayer local aggregation. Typically, for each node, one layer of a GNN aggregates information from its 1-hop neighborhood. Stacking $L$ such local aggregation layers allows each node to access information that is $L$ hops away. To be specific, the $\ell$-th layer of a $L$-layer GNN ($\ell = 1, 2, \ldots, L$) can be described as

$$z_v^{(\ell)} = \text{TRANSFORM}^{(\ell)} \left( \text{AGGREGATE}^{(\ell)} \left( \{ z_u^{(\ell-1)} : u \in \mathcal{N}_1(v) \cup v \} \right) \right) \in \mathbb{R}^f, \ \forall v \in V, \tag{3}$$

where $z_v^{(0)} = x_v$, and $z_v = z_v^{(L)}$ represents the local node embedding. The $\text{AGGREGATE}^{(\ell)}(\cdot)$ and $\text{TRANSFORM}^{(\ell)}(\cdot)$ functions represent the spatial aggregation and feature transformation step introduced in Section 2.1, respectively. With the above framework, GNNs can capture the node feature information from nodes within a local neighborhood as well as the structural information.

When either MLP or GNNs is used as the local embedding step, the local node embedding $z_v$ only contains local information of a node $v$. However, $z_v$ can be used to guide non-local aggregation, as distant but informative nodes are likely to have similar node features and local structures. Based on this intuition, we propose the attention-guided sorting to enable the non-local aggregation.

**Attention-Guided Sorting:** The basic idea of the attention-guided sorting is to learn an ordering of nodes, where distant but informative nodes are put near each other. Specifically, given the local node embedding $z_v$ obtained through the local embedding step, we compute one set of attention scores by

$$a_v = \text{ATTEND}(c, z_v) \in \mathbb{R}, \ \forall v \in V, \tag{4}$$

where $c$ is a calibration vector that is randomly initialized and jointly learned during training (Yang et al., 2016). In this attention operator, $c$ serves as the query vector and $z_v$ are the key vectors. In

addition, we also treat $z_v$ as the value vectors. However, unlike the attention mechanism introduced in Section 2.3, we use the attention scores to sort the value vectors instead of computing a weighted sum to aggregating them. Note that originally there is no ordering among nodes in a graph. To be specific, as $a_v$ and $z_v$ have one-to-one correspondence through Equation (4), sorting the attention scores in non-decreasing order into $(a_1, a_2, \ldots, a_n)$ provides an ordering among nodes, where $n = |V|$ is the number of nodes in the graph. The resulting sequence of local node embeddings can be denoted as $(z_1, z_2, \ldots, z_n)$.

The attention process in Equation (4) can be also understood as a projection of local node embeddings onto a 1-dimensional space. The projection depends on the concrete $\text{ATTEND}(\cdot)$ function and the calibration vector $c$. As indicated by its name, the calibration vector $c$ is used to calibrate the 1-dimensional space, in order to push distant but informative nodes close to each other in this space. This goal is fulfilled through the following non-local aggregation step and the training of the calibration vector $c$, as demonstrated below.

**Non-Local Aggregation:** We point out that, with the attention-guided sorting, the non-local aggregation can be achieved by convolution, the most common local aggregation operator in deep learning. Specifically, given the sorted sequence of local node embeddings $(z_1, z_2, \ldots, z_n)$, we compute

$$(\hat{z}_1, \hat{z}_2, \ldots, \hat{z}_n) = \text{CONV}(z_1, z_2, \ldots, z_n), \tag{5}$$

where the $\text{CONV}(\cdot)$ function represents a 1D convolution with appropriate padding. Note that the $\text{CONV}(\cdot)$ function can be replaced by a 1D convolutional neural network as long as the number of input and output vectors remains the same.

To see how the $\text{CONV}(\cdot)$ function performs non-local aggregation with the attention-guided sorting, we take an example where the $\text{CONV}(\cdot)$ function is a 1D convolution of kernel size $2s + 1$. In this case, $\hat{z}_i$ is computed from $(z_{i+s}, \ldots, z_{i-s})$, corresponding to the receptive field of the $\text{CONV}(\cdot)$ function. As a result, if the attention-guided sorting leads to $(z_{i+s}, \ldots, z_{i-s})$ containing nodes that are distant but informative to $z_i$, the output $\hat{z}_i$ aggregates non-local information. Another view is that we can consider the attention-guided sorting as re-connects nodes in the graph, where $(z_{i+s}, \ldots, z_{i-s})$ can be treated as the 1-hop neighborhood of $z_i$. After the $\text{CONV}(\cdot)$ function, $\hat{z}_i$ and $z_i$ are concatenated as the input to a classifier to predict the label of the corresponding node, where both non-local and local dependencies can be captured. In order to enable the end-to-end training of the calibration vector $c$, we modify Equation (5) into

$$(\hat{z}_1, \hat{z}_2, \ldots, \hat{z}_n) = \text{CONV}(a_1 z_1, a_2 z_2, \ldots, a_n z_n), \tag{6}$$

where we multiply the attention score with the corresponding local node embedding. As a result, the calibration vector $c$ receives gradients through the attention scores during training.

The remaining question is how to make sure that the attention-guided sorting pushes distant but informative nodes together. The short answer is that it is not necessary to guarantee this, as the requirement of non-local aggregation depends on the concrete graphs. In fact, our proposed framework grants GNNs the ability of non-local aggregation but lets the end-to-end training process determine whether to use non-local information. The back-propagation from the supervised loss will tune the calibration vector $c$ and encourage $\hat{z}_i$ to capture useful information that is not encoded by $z_i$. In the case of disassortative graphs, $\hat{z}_i$ usually needs to aggregate information from distant but informative nodes. Hence, the calibration vector $c$ tends to arrange the attention-guided sorting to put distant but informative nodes together, as demonstrated experimentally in Section 4.5. On the other hand, nodes within the local neighborhood are usually much more informative than distant nodes in assortative graphs. In this situation, $\hat{z}_i$ may simply perform local aggregation that is similar to GNNs.

In Section 4, we demonstrate the effectiveness of our proposed non-local aggregation framework on six disassortative graph datasets. In particular, we achieve the state-of-the-art performance on all the datasets with significant improvements over previous methods.

## 3.2 TIME COMPLEXITY ANALYSIS

We perform theoretical analysis of the time complexity of our proposed framework. As discussed in Section 2.3, using the attention mechanism (Vaswani et al., 2017; Wang et al., 2018; 2020) to achieve non-local aggregation requires $O(n^2)$ time for a graph of $n$ nodes. Essentially, the $O(n^2)$

time complexity is due to the fact that the ATTEND($\cdot$) function needs to be computed between every pair of nodes. In particular, the recently proposed Geom-GCN (Pei et al., 2020) contains a similar non-local aggregation step. For each $v \in V$, Geom-GCN finds the set of nodes from which the Euclidean distance to $v$ is less than a pre-defined number, where the Euclidean distance between every pair of nodes needs to be computed. As the computation of the the Euclidean distance between two nodes can be understood as the ATTEND($\cdot$) function, Geom-GCN has at least $O(n^2)$ time complexity.

In contrast, our proposed non-local aggregation framework requires only $O(n \log n)$ time. To see this, note that the ATTEND($\cdot$) function in Equation (4) only needs to be computed once, instead of iterating it for each node. As a result, computing the attention scores only takes $O(n)$ time. Therefore, the time complexity of sorting, *i.e.* $O(n \log n)$, dominates the total time complexity of our proposed framework. In Section 4.6, we compare the real running time on different datasets among common GNNs, Geom-GCN, and our non-local GNNs as introduced in the next section.

### 3.3 EFFICIENT NON-LOCAL GRAPH NEURAL NETWORKS

We apply our proposed non-local aggregation framework to build efficient non-local GNNs. Recall that our proposed framework starts with the local embedding step, followed by the attention-guided sorting and the non-local aggregation step.

In particular, the local embedding step can be implemented by either MLP or common GNNs, such as GCNs (Kipf & Welling, 2017) or GATs (Veličković et al., 2018). MLP extracts the local node embedding only from the node feature vector and excludes the information from nodes within the local neighborhood. This property can be helpful on some disassortative graphs, where nodes within the local neighborhood provide more noises than useful information. On other disassortative graphs, informative nodes locate in both local neighborhood and distant locations. In this case, GNNs are more suitable as the local embedding step. Depending on the disassortative graphs in hand, we build different non-local GNNs with either MLP or GNNs as the local embedding step. In Section 4.3, we show that these two categories of disassortative graphs can be distinguished through simple experiments, where we apply different non-local GNNs accordingly. Specifically, the number of layers is set to 2 for both MLP and GNNs.

In terms of the attention-guided sorting, we only need to specify the ATTEND($\cdot$) function in Equation (4). In order to make it as efficient as possible, we choose the ATTEND($\cdot$) function as

$$a_v = \text{ATTEND}(c, z_v) = c^T z_v \in \mathbb{R}, \ \forall v \in V, \tag{7}$$

where $c$ is part of the training parameters, as described in Section 3.1.

With the attention-guided sorting, we can implement the non-local aggregation step through convolution, as explained in Section 3.1 and shown in Equation (6). Specifically, CONV($\cdot$) function is set as a 2-layer convolutional neural network composed of two 1D convolutions. The kernel size is set to 3 or 5 depending on the datasets. The activation function is ReLU (Krizhevsky et al., 2012).

Finally, we use a linear classifier that takes the concatenation of $\hat{z}_i$ and $z_i$ as inputs and makes prediction for the corresponding node. Depending on the local embedding step, we build three efficient non-local GNNs, namely non-local MLP (NLMLP), non-local GCN (NLGCN), and non-local GAT (NLGAT). The models can be end-to-end trained with the classification loss.

## 4 EXPERIMENTS

### 4.1 DATASETS

We perform experiments on six disassortative graph datasets (Rozemberczki et al., 2019; Tang et al., 2009; Pei et al., 2020) (*Chameleon*, *Squirrel*, *Actor*, *Cornell*, *Texas*, *Wisconsin*) and three assortative graph datasets (Sen et al., 2008) (*Cora*, *Citeseer*, *Pubmed*). These datasets are commonly used to evaluate GNNs on node classification tasks (Kipf & Welling, 2017; Veličković et al., 2018; Gao et al., 2018; Pei et al., 2020). We provide detailed descriptions of disassortative graph datasets in Appendix A.1. In order to distinguish assortative and disassortative graph datasets, Pei et al. (2020)

propose a metric to measure the homophily of a graph $\mathcal{G}$, defined as

$$H(\mathcal{G}) = \frac{1}{|V|} \sum_{v \in V} \frac{\text{Number of } v\text{'s directly connected nodes who have the same label as } v}{\text{Number of } v\text{'s directly connected nodes}}. \quad (8)$$

Intuitively, a large $H(\mathcal{G})$ indicates an assortative graph, and vice versa. The $H(\mathcal{G})$ and other statistics are summarized in Table 1.

Table 1: Statistics of the nine datasets used in our experiments. The definition of $H(\mathcal{G})$ is provided in Section 4.1. $H(\mathcal{G})$ can be used to distinguish assortative and disassortative graph datasets.

| Datasets | Assortative | | | Disassortative | | | | | |
|---|---|---|---|---|---|---|---|---|---|
| | *Cora* | *Citeseer* | *Pubmed* | *Chameleon* | *Squirrel* | *Actor* | *Cornell* | *Texas* | *Wisconsin* |
| $H(\mathcal{G})$ | 0.83 | 0.71 | 0.79 | 0.25 | 0.22 | 0.24 | 0.11 | 0.06 | 0.16 |
| #Nodes | 2708 | 3327 | 19717 | 2277 | 5201 | 7600 | 183 | 183 | 251 |
| #Edges | 5429 | 4732 | 44338 | 36101 | 217073 | 33544 | 295 | 309 | 499 |
| #Features | 1433 | 3703 | 500 | 2325 | 2089 | 931 | 1703 | 1703 | 1703 |
| #Classes | 7 | 6 | 3 | 5 | 5 | 5 | 5 | 5 | 5 |

In our experiments, we focus on comparing the model performance on disassortative graph datasets, in order to demonstrate the effectiveness of our non-local aggregation framework. The performances on assortative graph datasets are provided for reference, indicating that the proposed framework will not hurt the performance when non-local aggregation is not strongly desired.

## 4.2 BASELINES

We compare our proposed non-local MLP (NLMLP), non-local GCN (NLGCN), and non-local GAT (NLGAT) with various baselines: (1) MLP is the simplest deep learning model. It makes prediction solely based on the node feature vectors, without aggregating any local or non-local information. (2) GCN (Kipf & Welling, 2017) and GAT (Veličković et al., 2018) are the most common GNNs. As introduced in Section 2.1, they only perform local aggregation. (3) Geom-GCN (Pei et al., 2020) is a recently proposed GNN that can capture long-range dependencies. It is the current state-of-the-art model on several disassortative graph datasets. Geom-GCN requires the use of different node embedding methods, such as Isomap (Tenenbaum et al., 2000), Poincare (Nickel & Kiela, 2017), and struc2vec (Ribeiro et al., 2017). We simply report the best results from Pei et al. (2020) for Geom-GCN and the following two variants without specifying the node embedding method. (4) Geom-GCN-g (Pei et al., 2020) is a variant of Geom-GCN that performs local aggregation only. It is similar to common GNNs. (5) Geom-GCN-s (Pei et al., 2020) is a variant of Geom-GCN that does not force local aggregation. The designed functionality is similar to our NLMLP.

We implement MLP, GCN, GAT, and our methods using Pytorch (Adam et al., 2017) and Pytorch Geometric (Fey & Lenssen, 2019). As has been discussed[1], in fair settings, the results of GCN and GAT differ from those in Pei et al. (2020).

On each dataset, we follow Pei et al. (2020) and randomly split nodes of each class into 60%, 20%, and 20% for training, validation, and testing. The experiments are repeatedly run 10 times with different random splits and the average test accuracy over these 10 runs are reported. Testing is performed when validation accuracy achieves maximum on each run. Apart from the details specified in Section 3.3, we tune the following hyperparameters individually for our proposed models: (1) the number of hidden unit $\in \{16, 48, 96\}$, (2) dropout rate $\in \{0, 0.5, 0.8\}$, (3) weight decay $\in \{0, 5e\text{-}4, 5e\text{-}5, 5e\text{-}6\}$, and (4) learning rate $\in \{0.01, 0.05\}$.

## 4.3 ANALYSIS OF DISASSORTATIVE GRAPH DATASETS

As discussed in Section 3.3, the disassortative graph datasets can be divided into two categories. Nodes within the local neighborhood provide more noises than useful information in disassortative graphs belonging to the first category. Therefore, local aggregation should be avoided in models on such disassortative graphs. As for the second category, informative nodes locate in both local neighborhood and distant locations. Intuitively, a graph with lower $H(\mathcal{G})$ is more likely to be in the first category. However, it is not an accurate way to determine the two categories.

---

[1]https://openreview.net/forum?id=S1e2agrFvS&noteId=8tGKV1oSzCr

Knowing the exact category of a disassortative graph is crucial, as we need to apply non-local GNNs accordingly. As analyzed above, the key difference lies in whether the local aggregation is useful. Hence, we can distinguish two categories of disassortative graphs by comparing the performance between MLP and common GNNs (GCN, GAT) on each of the six disassortative graph datasets.

Table 2: Comparisons between MLP and common GNNs. These analytical experiments are used to determine the two categories of disassortative graph datasets, as introduced in Section 4.3.

| | Assortative | | | Disassortative | | | | | |
|---|---|---|---|---|---|---|---|---|---|
| **Datasets** | *Cora* | *Citeseer* | *Pubmed* | *Chameleon* | *Squirrel* | *Actor* | *Cornell* | *Texas* | *Wisconsin* |
| MLP | $76.5_{\pm1.3}$ | $73.6_{\pm1.9}$ | $87.5_{\pm0.4}$ | $48.5_{\pm3.0}$ | $31.5_{\pm1.4}$ | $\mathbf{35.1}_{\pm0.8}$ | $\mathbf{81.6}_{\pm6.3}$ | $\mathbf{81.3}_{\pm7.1}$ | $\mathbf{84.9}_{\pm5.3}$ |
| GCN | $88.2_{\pm1.2}$ | $75.7_{\pm1.3}$ | $\mathbf{88.4}_{\pm0.6}$ | $\mathbf{67.6}_{\pm2.4}$ | $\mathbf{54.9}_{\pm1.9}$ | $30.3_{\pm1.6}$ | $54.2_{\pm7.3}$ | $61.1_{\pm7.0}$ | $59.6_{\pm4.5}$ |
| GAT | $\mathbf{88.4}_{\pm1.4}$ | $\mathbf{76.1}_{\pm1.0}$ | $87.0_{\pm0.3}$ | $65.0_{\pm3.7}$ | $51.3_{\pm2.5}$ | $29.4_{\pm1.2}$ | $56.3_{\pm4.3}$ | $57.9_{\pm6.1}$ | $57.8_{\pm4.3}$ |

The results are summarized in Table 2. We can see that *Actor*, *Cornell*, *Texas*, and *Wisconsin* fall into the first category, while *Chameleon* and *Squirrel* belong to the second category. We add the performance on assortative graph datasets for reference, where the local aggregation is effective so that GNNs tend to outperform MLP.

## 4.4    COMPARISONS WITH BASELINES

According to the insights from Section 4.3, we apply different non-local GNNs according to the category of disassortative graph datasets, and make comparisons with corresponding baselines.

Specifically, we employ NLMLP on *Actor*, *Cornell*, *Texas*, and *Wisconsin*. The corresponding baselines are MLP, Geom-GCN, and Geom-GCN-s, as Table 2 has shown that GCN and GAT perform much worse than MLP on these datasets. And Geom-GCN-g is similar to GCN and has worse performance than Geom-GCN-s, which is shown in Ap-

Table 3: Comparisons between our NLMLP and strong baselines on the four disassortative graph datasets belonging to the first category as defined in Section 4.3.

| **Datasets** | *Actor* | *Cornell* | *Texas* | *Wisconsin* |
|---|---|---|---|---|
| MLP | $35.1_{\pm0.8}$ | $81.6_{\pm6.3}$ | $81.3_{\pm7.1}$ | $84.9_{\pm5.3}$ |
| Geom-GCN | 31.6 | 60.8 | 67.6 | 64.1 |
| Geom-GCN-s | 34.6 | 75.4 | 73.5 | 80.4 |
| NLMLP | $\mathbf{37.9}_{\pm1.3}$ | $\mathbf{84.9}_{\pm5.7}$ | $\mathbf{85.4}_{\pm3.8}$ | $\mathbf{87.3}_{\pm4.3}$ |

pendix A.2. The comparison results are reported in Table 3. While Geom-GCN-s are the previous state-of-the-art GNNs on these datasets (Pei et al., 2020), we find that MLP consistently outperforms Geom-GCN-s by large margins. In particular, although Geom-GCN-s does not explicitly perform local aggregation, it is still outperformed by MLP. A possible explanation is that Geom-GCN-s uses pre-trained node embeddings, which aggregates information from the local neighborhood implicitly. In contrast, our NLMLP is built upon MLP with the proposed non-local aggregation framework, which excludes the local noises and collects useful information from non-local informative nodes. The NLMLP sets the new state-of-the-art performance on these disassortative graph datasets.

On *Chameleon* and *Squirrel* that belong to the second category of disassortative graph datasets, we apply NL-GCN and NLGAT accordingly. The baselines are GCN, GAT, Geom-GCN, and Geom-GCN-g. On these datasets, these baselines that explicitly perform local aggregation show advantages over MLP and Geom-GCN-s, as shown in Appendix A.2. As shown in Table 4, our proposed NLGCN achieves the best performance on both datasets. In addition, it is worth noting that our NLGCN and NL-GAT are built upon GCN and GAT, respectively. They show improvements over their counterparts, which indicates that the advantages of our proposed non-local aggregation framework are general for common GNNs.

Table 4: Comparisons between our NL-GCN, NLGAT and strong baselines on the two disassortative graph datasets belonging to the second category as defined in Section 4.3.

| **Datasets** | *Chameleon* | *Squirrel* |
|---|---|---|
| GCN | $67.6_{\pm2.4}$ | $54.9_{\pm1.9}$ |
| GAT | $65.0_{\pm3.7}$ | $51.3_{\pm2.5}$ |
| Geom-GCN | 60.9 | 38.1 |
| Geom-GCN-g | 68.0 | 46.0 |
| NLGCN | $\mathbf{70.1}_{\pm2.9}$ | $\mathbf{59.0}_{\pm1.2}$ |
| NLGAT | $65.7_{\pm1.4}$ | $56.8_{\pm2.5}$ |

We provide the results of all the models on all datasets in Appendix A.2 for reference.

## 4.5    ANALYSIS OF THE ATTENTION-GUIDED SORTING

We analyze the results of the attention-guided sorting in our proposed framework, in order to show that our non-local GNNs indeed perform non-local aggregation.

Suppose the attention-guided sorting leads to the sorted sequence $(z_1, z_2, \ldots, z_n)$, which goes through a convolution or CNN into $(\hat{z}_1, \hat{z}_2, \ldots, \hat{z}_n)$. As discussed in Section 3.1, we can consider the sequence $(z_1, z_2, \ldots, z_n)$ as a re-connected graph $\hat{\mathcal{G}}$, where we treat nodes within the receptive field of $\hat{z}_i$ as directly connected to $z_i$, *i.e.* $z_i$'s 1-hop neighborhood. The information within this new 1-hop neighborhood will be aggregated. If our non-local GNNs indeed perform non-local aggregation, the homophily of the re-connected graph should be larger than the original graph. Therefore, we compute $H(\hat{\mathcal{G}})$ for each dataset to verify this statement. Following Section 4.4, we apply NLMLP on *Actor*, *Cornell*, *Texas*, and *Wisconsin* and NLGCN on *Chameleon* and *Squirrel*.

Figure 1 compares $H(\hat{\mathcal{G}})$ with $H(\mathcal{G})$ for each dataset. We can observe that $H(\hat{\mathcal{G}})$ is much larger than $H(\mathcal{G})$, indicating that distant but informative nodes are near each other in the re-connected graph $\hat{\mathcal{G}}$. We also provide the visualizations of the sorted sequence for *Cornell* and *Texas*. We can see that nodes with the same label tend to be clustered together. These facts indicate that our non-local GNNs perform non-local aggregation with the attention-guided sorting.

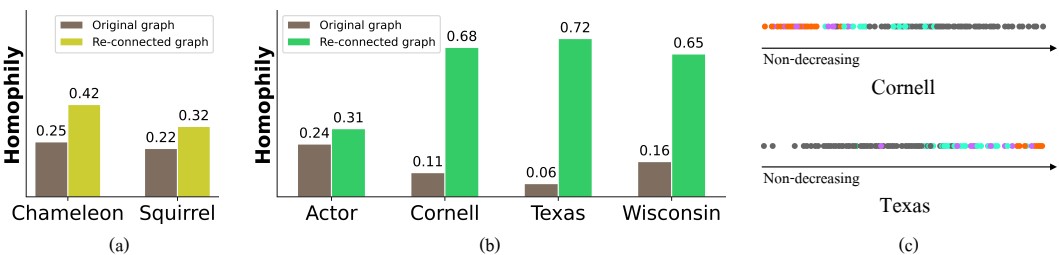

Figure 1: (a) Comparisons of the homophily between the original graph and the re-connected graph given by our NLGCN on *Chameleon* and *Squirrel*. (b) Comparisons of the homophily between the original graph and the re-connected graph given by our NLMLP on *Actor*, *Cornell*, *Texas*, and *Wisconsin*. (c) Visualization of sorted node sequence after the attention-guided sorting for *Cornell* and *Texas*. The colors denote node labels. Details are explained in Section 4.5.

## 4.6 EFFICIENCY COMPARISONS

As analyzed in Section 3.2, our proposed non-local aggregation framework is more efficient than previous methods based on the original attention mechanism, such as Geom-GCN (Pei et al., 2020). Concretely, our method requires only $O(n \log n)$ computation time in contrast to $O(n^2)$. In this section, we compare the real running time to verify our analysis. Specifically, we compare NLGCN with Geom-GCN as well as GCN and GAT. For Geom-GCN, we use the code provided in Pei et al. (2020). Each model is trained for 500 epochs on each dataset and the average training time per epoch is reported.

Table 5: Comparisons in terms of real running time (*milliseconds*).

|  | Chameleon | Squirrel |
| --- | --- | --- |
| GCN | 22.2 (1.0×) | 14.3 (1.0×) |
| GAT | 33.2 (1.5×) | 163.3 (11.4×) |
| Geom-GCN | 3615.0 (163.1×) | 10430.0 (727.3×) |
| NLGCN | 26.3 (1.2×) | 39.6 (2.8×) |

The results are shown in Table 5. Although our NLGCN is built upon GCN, it is just slightly slower than GCN and faster than GAT, showing the efficiency of our non-local aggregation framework. On the other hand, Geom-GCN is significantly slower due to the fact that it has $O(n^2)$ time complexity.

## 5 CONCLUSION

In this work, we propose a simple yet effective non-local aggregation framework for GNNs. The core of the framework is an efficient attention-guided sorting, which enables non-local aggregation through convolution. The proposed framework can be easily used to build non-local GNNs with low computational costs. We perform thorough experiments on node classification tasks to evaluate our proposed method. In particular, we experimentally analyze existing disassortative graph datasets and apply different non-local GNNs accordingly. The results show that our non-local GNNs significantly outperform previous state-of-the-art methods on six benchmark datasets of disassortative graphs, in terms of both accuracy and speed.

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

# A APPENDIX

## A.1 DETAILS OF DISASSORTATIVE GRAPH DATASETS

Here are the details of disassortative graph datasets used in our experiments:

- *Chameleon* and *Squirrel* are Wikipedia networks (Rozemberczki et al., 2019) where nodes represent web pages from Wikipedia and edges indicate mutual links between pages. Node feature vectors are bag-of-word representation of informative nouns in the corresponding pages. Each node is labeled with one of five classes according to the number of the average monthly traffic of the web page.

- *Actor* is an actor co-occurrence network, where nodes denote actors and edges indicate co-occurrence on the same web page from Wikipedia. It is extracted from the film-director-actor-writer network proposed by Tang et al. (Tang et al., 2009). Node feature vectors are bag-of-word representation of keywords in the actors' Wikipedia pages. Each node is labeled with one of five classes according to the topic of the actor's Wikipedia page.

- *Cornell*, *Texas*, and *Wisconsin* come from the WebKB dataset collected by Carnegie Mellon University. Nodes represent web pages and edges denote hyperlinks between them. Node feature vectors are bag-of-word representation of the corresponding web pages. Each node is labeled with one of student, project, course, staff, and faculty.

## A.2 FULL EXPERIMENTAL RESULTS

Table 6: Comparisons between our NLMLP, NLGCN, NLGAT and baselines on all the nine datasets.

| Datasets | Assortative | | | Disassortative | | | | | |
|---|---|---|---|---|---|---|---|---|---|
| | *Cora* | *Citeseer* | *Pubmed* | *Chameleon* | *Squirrel* | *Actor* | *Cornell* | *Texas* | *Wisconsin* |
| MLP | $76.5_{\pm1.3}$ | $73.6_{\pm1.9}$ | $87.5_{\pm0.4}$ | $48.5_{\pm3.0}$ | $31.5_{\pm1.4}$ | $35.1_{\pm0.8}$ | $81.6_{\pm6.3}$ | $81.3_{\pm7.1}$ | $84.9_{\pm5.3}$ |
| GCN | $88.2_{\pm1.2}$ | $75.7_{\pm1.3}$ | $88.4_{\pm0.6}$ | $67.6_{\pm2.4}$ | $54.9_{\pm1.9}$ | $30.3_{\pm1.6}$ | $54.2_{\pm7.3}$ | $61.1_{\pm7.0}$ | $59.6_{\pm4.5}$ |
| GAT | $88.4_{\pm1.4}$ | $76.1_{\pm1.0}$ | $87.0_{\pm0.3}$ | $65.0_{\pm3.7}$ | $51.3_{\pm2.5}$ | $29.4_{\pm1.2}$ | $56.3_{\pm4.3}$ | $57.9_{\pm6.1}$ | $57.8_{\pm4.3}$ |
| Geom-GCN | 85.3 | 78.0 | 90.1 | 60.9 | 38.1 | 31.6 | 60.8 | 67.6 | 64.1 |
| Geom-GCN-g | 87.0 | **80.6** | **90.7** | 68.0 | 46.0 | 32.0 | 67.0 | 73.1 | 69.4 |
| Geom-GCN-s | 73.3 | 72.2 | 87.0 | 61.6 | 38.0 | 34.6 | 75.4 | 73.5 | 80.4 |
| NLMLP | $76.9_{\pm1.8}$ | $73.4_{\pm1.9}$ | $88.2_{\pm0.5}$ | $50.7_{\pm2.2}$ | $33.7_{\pm1.5}$ | $\mathbf{37.9}_{\pm1.3}$ | $\mathbf{84.9}_{\pm5.7}$ | $\mathbf{85.4}_{\pm3.8}$ | $\mathbf{87.3}_{\pm4.3}$ |
| NLGCN | $88.1_{\pm1.0}$ | $75.2_{\pm1.4}$ | $89.0_{\pm0.5}$ | $\mathbf{70.1}_{\pm2.9}$ | $\mathbf{59.0}_{\pm1.2}$ | $31.6_{\pm1.0}$ | $57.6_{\pm5.5}$ | $65.5_{\pm6.6}$ | $60.2_{\pm5.3}$ |
| NLGAT | $\mathbf{88.5}_{\pm1.8}$ | $76.2_{\pm1.6}$ | $88.2_{\pm0.3}$ | $65.7_{\pm1.4}$ | $56.8_{\pm2.5}$ | $29.5_{\pm1.3}$ | $54.7_{\pm7.6}$ | $62.6_{\pm7.1}$ | $56.9_{\pm7.3}$ |

