# OpenReview forum: "Non-Local Graph Neural Networks"
_ICLR.cc/2021/Conference — Reject_

### Official Review · AnonReviewer2 · 2020-10-26
**the idea of “push” the distant but informative nodes together is not reflected**

**Rating:** 6
**Confidence:** 5

**Review:**

This paper targets on addressing the node embedding problem in disassortative graphs. A non-local aggregation framework is proposed, since local aggregation may be harmful for some disassortative graphs. To address the high computational cost in the recent Geom-GCN model that has an attention-like step to compute the Euclidean distance between every pair of nodes, an idea of attention-guided sorting is introduced. It learns an ordering of nodes, such that distant but informative nodes are put near each other. The sorting order depends on the attention scores computed with the local embedding vector of a node. Then Covn(.) function is applied on the sorted sequence of local node embeddings to obtain the non-local embedding. The final node embedding is then the concatenation of the local and non-local embedding, which is used for node classification.

The presented simple approach is an interesting idea to “push” the distant but informative nodes together. However, it is unclear how the “attention-guided sorting” is aware of the “distant” nodes. The local node embedding vectors z can be obtained either by the node content, or by GNN. If z is from the node content only, the attention score a is calculated without consideration how nodes are close or distant on the graph. The whole approach works purely for node content classification. If z is from GNN,  nodes close on the graph have similar z embedding vectors and thus will be sorted next to each other. Then, the sorting doesn’t take “distant” nodes close.

Although the experimental results show the proposed approach performs better than several baselines, more and stronger GNN models are expected to be compared with, e.g., GINs. Especially on Chameleon and Squirrel datasets, theses two “disassortative” graphs can be handled by GNN kinds of models. The node classification in other four “disassortative” graphs in fact can be treated as a standard class classification task by ignoring the graph structures, as MLP on node features is already good.

Thanks for the clarifications from the authors. The discussion was very helpful.

---

> ### Author Response · Authors · 2020-11-14
> **Our response to Reviewer #2**
>
> Thank you for the helpful comments. We address the concerns as follows.
>
> Q1: The presented simple approach is an interesting idea to “push” the distant but informative nodes together. However, it is unclear how the “attention-guided sorting” is aware of the “distant” nodes. The local node embedding vectors z can be obtained either by the node content, or by GNN. If z is from the node content only, the attention score a is calculated without consideration how nodes are close or distant on the graph. The whole approach works purely for node content classification. If z is from GNN, nodes close on the graph have similar z embedding vectors and thus will be sorted next to each other. Then, the sorting doesn’t take “distant” nodes close.
>
> A1:
>
> Intuitively, if $z$ is from the node content only, nodes with similar embedding $z$ will obtain similar attention scores. Then, these nodes can be further aggregated by our subsequent 1-D convolution. This aggregation makes the feature vectors of nodes in the same class to be similar with each other, thus easing the following classification. That’s why our NLMLP can outperform MLP, as show in Table 3 in our paper. If $z$ is derived by local GNN, nodes close on graph would have similar $z$ embedding. However, informative distant nodes might also have similar $z$ embeddings since this is a disassortative graph. Thus, such long-range dependencies can be captured by the subsequent attention-guided sorting and 1-D convolution.  This has been verified by the improvement shown in Table 4.
>
> Empirically, the visualization of homophily in figure 1 demonstrates that our non-local GNN indeed perform desired non-local aggregation on disassortative graphs.
>
> Q2: Although the experimental results show the proposed approach performs better than several baselines, more and stronger GNN models are expected to be compared with, e.g., GINs. Especially on Chameleon and Squirrel datasets, theses two “disassortative” graphs can be handled by GNN kinds of models. The node classification in other four “disassortative” graphs in fact can be treated as a standard class classification task by ignoring the graph structures, as MLP on node features is already good.
>
> A2:
>
> (1) We didn't include the GIN as baselines for the following reason. GIN is proved to be powerful for distinguishing different graph structures, and it shines on graph-level tasks. However, as shown empirically in previous works, such as [1], GIN usually performs not as good as GCN on node classification tasks. To make it more convincing, we add the result of GIN on Chameleon and Squirrel for reference, as shown below.
>
>  |           |Chameleon|    Squirrel  |
>
> |GIN     |   66.6+/-3.0|  50.9+/-3.4|
>
> |NLGCN|  70.1+/-2.9|   59.0+/-1.2|
>
> (2) As shown in our paper, MLP actually performs much better than local GNNs on several disassortative graphs. This shows local aggregation which adopted by most GNNs hurts the performance, and the community should explore more effective methods for disassortative graphs. Our non-local GNN is an early attempt for such goal and has been shown to be effective to some degree. Most of the existing graph learning methods are based on the assumption of high homophily. We argue that the information contained in the disassortative graphs is also valuable and deserves more investigation and attention from the community.
>
> Thank you again for your insightful and constructive review. We hope that we addressed you concerns.
>
> [1] Wang et el.. Demystifying graph neural network via graph filter assessment. 2019.

---

> > ### Comment · AnonReviewer2 · 2020-11-24
> > **Thanks for the reply**
> >
> > Thanks to the authors for their reply.
> >
> > In A1, "informative distant nodes might also have similar  embeddings since this is a disassortative graph". Is there any empirically analysis in one or two disassortative graphs, showing how much "informative distant nodes might also have similar  embedding"? Fig.1(c) shows the sorted nodes with different labels. It would be very interesting to see how much "informative distant nodes" were pushed close to together.

---

> > > ### Author Response · Authors · 2020-11-24
> > > **Authors' response (2nd round)**
> > >
> > > Thank you for your kind response.
> > >
> > > Great point! Investigating how much "informative distant nodes" were pushed close is very interesting and can demonstrate the effectiveness of the proposed non-local framework. Actually, we considered and illustrated this point empirically in Section 4.5 and Figure 1 (a)&(b), as shown in our paper. To make it clearer, we would like to explain it intuitively to the reviewer.
> > >
> > > After applying the attention-guided sorting, we obtain a sorted sequence of nodes. Then, we apply a 1-D CNN on this sorted sequence to achieve non-local aggregation. Hence, we can consider the sequence as a re-connected graph, where for each node $v$, we can treat the nodes within the receptive field (of the 1-D CNN) of $v$ as directly connected to $v$, *i.e.*, 1-hop neighborhood. Then, the information within this new 1-hop neighborhood can be aggregated by the 1-D CNN. Following this idea, if our attention-guided sorting actually pushes distant but informative nodes together, then the homophily of the re-connected graph should be larger than the original graph. As shown in Figure 1 (a)&(b), the homophily of the re-connected graphs is much larger than the original disassortative graphs. This indicates that our non-local GNNs indeed perform non-local aggregation with the attention-guided sorting.
> > >
> > > Thank you again for proposing this interesting and insightful comment. Hope we addressed your concerns.

---

### Official Review · AnonReviewer1 · 2020-10-28
**Review of Non-Local Graph Neural Networks**

**Rating:** 4
**Confidence:** 4

**Review:**

Summary:
The goal of the paper is to perform node classification for graphs. The authors propose a strategy to augment message passing graph neural networks with information from non-local nodes in the graph - with a focus on dis-assortative graphs. Dis-assortative graphs are graph datasets - where nodes with identical node labels are distant from each other in terms of edge connectivity.

With node representation learnt from standard graph neural networks, etc., the authors propose to use an attention guided sorting mechanism, to create a proxy graph, where nodes which may have identical node labels be connected to each other (analogous to creating a k-nearest neighbor graph). Message passing is then employed on the proxy graph to learn final representations for the nodes.  Since the authors employ a single vector, namely 'c' (which they call calibration vector), to capture the 'importance of information' shared across different nodes  - there is a speedup in comparison to strategies which employ a pairwise comparison between all nodes in the graph.

Pros:
1. The idea to create a proxy graph to capture non local information is interesting
2. The proposed technique can be augmented with almost any existing GNN

My Concerns:
1. (Dis-assortative or i.i.d.): - The authors in Figure 1 - show that homophily of the created proxy graph is a value larger than that of the original graph. However, from table A.2 in the appendix - it is clear to see that MLP's outperform GNN's with or without the attention sorting in the dis-assortative graphs - and the performance of the MLP's and proposed augmented NLMLP are well within one standard deviation from each other. This questions the need to employ a proxy graph construction on top of MLP's for these datasets as it appears like the data can be treated as i.i.d (and not relational). Moreover, these datasets (used from Pei et al. 2020) are extremely small to draw any significant conclusion. Also almost no gains are seen on the assortative datasets Citeseer, Cora, Pubmed (Please add datasets from OGB) - and their running times (when augmented with a proxy graph - are the gains worthy of increased run times?).
2. (Baselines): Since the authors propose a strategy to construct a proxy graph (and the number of neighbors of each node in the proxy graph is the same??) - baselines such as creating graphs where nodes with identical labels are connected are also connected to each other / GNN on simple k-Nearest neighbors created using initial features (While a simple k-NN might appear more expensive - but the computation here is a single time effort) appears crucial. Also add a baseline, where adjacency structure of the graphs are iteratively updated during training such as - Learning discrete structures for graph neural networks (Franceschi, et al. ICML 2019)
3. (Sufficiency, lack of details): The use of a single calibration vector may not be sufficient to sort the nodes - there are no guarantees in the paper to say when a single calibration vector would suffice. Also, the number of classes of nodes in each of the datasets used here are very small - and also does not trivially extend to multi-label classification of nodes. Also how do you also determine the number of neighbors in a proxy graph and do all nodes need to have the same number of neighbors in the proxy graph??? There are missing equations about how the calibration vector 'c' is learnt (what are the objective, etc) and the effect of running time when there are a large number of neighbors considered in the proxy graph -  without any equations its hard to argue against the case that the number of gradients to be computed would explode, when the number of neighbors are increased in the proxy graph (especially when jointly learning the GNN and the proposed augmentation).


Other minor concerns:
If possible, please include the difference between assortative and non-assortative graphs in the introduction - it makes it easier for a reader.

If details are added and the concerns are addressed, I will be happy to improve my score.

---

> ### Author Response · Authors · 2020-11-14
> **Our response to Reviewer #1 (Part 2)**
>
>
> Q3: (Sufficiency, lack of details): The use of a single calibration vector may not be sufficient to sort the nodes - there are no guarantees in the paper to say when a single calibration vector would suffice. Also, the number of classes of nodes in each of the datasets used here are very small - and also does not trivially extend to multi-label classification of nodes. Also how do you determine the number of neighbors in a proxy graph and do all nodes need to have the same number of neighbors in the proxy graph??? There are missing equations about how the calibration vector 'c' is learnt (what are the objective, etc) and the effect of running time when there are a large number of neighbors considered in the proxy graph - without any equations its hard to argue against the case that the number of gradients to be computed would explode, when the number of neighbors are increased in the proxy graph (especially when jointly learning the GNN and the proposed augmentation).
>
> A3:
> (1) Great point! We use a single calibration vector in our model just because one single calibration vector performs good and increasing the number of calibration vectors does not bring obvious gains on these available datasets. Yes, our attention-guided sorting step can also be augmented with multi-head versions, like the general attention mechanism. Hence, for the mentioned multi-label classification problem (although this is out of the scope of our paper), we might use multi-head classifier with multi-head attention-guided sorting to make multi-label predictions.
>
> (2) the number of neighbors in a “proxy graph” is like a normal hyperparameter. We tune it based on the validation set, as other hyperparameters. Particularly, the hyperparameter is set to $3$ or $5$. If we want to consider a large neighborhood in a “proxy graph”, we can stack more 1-D convolutional layers. This is identical to the general deep convolutional network, which is widely used and computationally affordable. Hence, we don’t need to worry about the inexistent exploding computation issue.
>
> (3) The calibration vector 'c' is learnt by receiving the gradient from Equation $6$, as described in the paragraph under that equation.
>
> Q4: Other minor concerns: If possible, please include the difference between assortative and non-assortative graphs in the introduction - it makes it easier for a reader.
>
>
> A4: Thank you for this helpful suggestion. We will add this clarification in the introducing in our revised version.
>
>
> Thank you again for these insightful and thorough reviews. Hope we addressed your concerns.

---

> > ### Comment · AnonReviewer1 · 2020-11-21
> > **Reply to response from authors**
> >
> > Thank you for responding to my concerns.
> > With regard to the dis-assortative vs iid: Thank you for adding the p-values.
> >
> > With regard to the other questions:
> > 1. It is still concerning that the number of neighbors in a proxy graph is fixed - and this number is pre-determined or is a hyper-parameter. Since the graphs tested here are all of very small size - I believe this cannot shed much light on how the value if this parameter is chosen.  I wanted to see the equation for the calibration vector update with the gradients - because this is going to be impacted by the number of neighbors in the proxy graph - chosen as the parameter - while - 3-5 for this parameter is okay for small graphs - it may not be for real world graphs and can cause computation issues.
> > 2. With regard to the baseline - is there a reason you had used GCN(after the KNN) in comparison to some-other graph neural network (asking because GCN is now 4 years old)?
> > 3. You mention the new baseline with the KNN after the MLP layer? Is the KNN based on the MLP output (and re-computed in every forward pass) or some-other way? How do you back-propagate through your selection - straight through estimator?
> > 4. It is a still a concern that there is no theoretical backing - on how an attention mechanism is going to push the distant nodes closer or just how one calibration vector is going to suffice (given that all graphs are small).
> >
> > Thanks,
> > Reviewer 1

---

> > > ### Author Response · Authors · 2020-11-22
> > > **Response to Reviewer #1 (2nd round)**
> > >
> > > Thank you for the kind response and helpful comments. We answer the remaining concerns as follows.
> > >
> > > Q1: We don't repeat the question since the space is limited :)
> > >
> > > A1:
> > >
> > > Yes, the number of neighbors in the “proxy” graph is a tuned hyperparameter and identical for all nodes in the graph. The key point is that this calibration vector will receive gradients from all the nodes according to Eq. (6) and (7) during back propagation, no matter what is the number of neighbors in the “proxy” graph. Hence, the computation cost for updating this calibration vector is similar to the one of updating a transformation parameter matrix (e.g., with shape $d \times 1$) (which is also applies to all nodes) in GCN, which is not expensive. If we want to consider larger neighborhoods in the “proxy” graph, we only need to increase the kernel size of the 1-D convolution. This will bring more computation to the CNN part, which is known to be inevitable if we want obtain larger receptive fields (on images or graphs). However, the computation about updating calibration vector keeps almost unchanged because it still receives gradients from all nodes.
> > >
> > > Overall, compared with GCN, the additional computation costs of non-local operations are mainly from the process of determining effective “non-local neighbors". In previous work Geom-GCN, since computing Euclidean distance between every pair of nodes is utilized, the computational complexity is $O(n^2)$. In contrast, our proposed non-local aggregation framework requires only $O(n\log n)$ time, dominated by the complexity of sorting. This is much more efficient especially when the number of nodes is large, as shown in Tab. 4 in our paper. Usually, an efficient implementation of GCN requires $O(nmd)$ computational cost, where $m$ is the number of edges and $d$ is the number of feature dimensions. Hence, our non-local aggregation usually has the computational cost of a similar order of magnitude, compared with GCN.
> > >
> > > A2: Yes, we choose GCN for two main reasons.
> > >
> > > (1)	For local aggregation, GCN performs better on these datasets among several popular GNNs. Although GCN is proposed four years ago, it is still a strong baseline. Specifically, GCN performs better than GAT and GIN on these datasets, as shown in the experiments in our paper and our response to Reviewer #2.
> > >
> > > (2)	In this baseline, GCN is only used for aggregating after KNN. We use it because it is the most representative aggregation operation for graph data.
> > >
> > > Q3: You mention the new baseline with the KNN after the MLP layer? Is the KNN based on the MLP output (and re-computed in every forward pass) or some-other way? How do you back-propagate through your selection - straight through estimator?
> > >
> > > A3:
> > > Yes, the KNN is based on the MLP output and re-computed in every forward pass. The backpropagation through KNN is similar to the backpropagation through MaxPooling operation since there are no trainable parameters in KNN. In this way, the MLP can be trained.
> > >
> > > Q4: It is a still a concern that there is no theoretical backing - on how an attention mechanism is going to push the distant nodes closer or just how one calibration vector is going to suffice (given that all graphs are small).
> > >
> > > A4:
> > > We need to point out that it is very challenging to provide strict theoretical support currently since training a good task-specific sorting is a widely existing challenge in deep learning.
> > >
> > > We provide an intuitive perspective to understand our approach as follows.
> > >
> > > You can understand this by comparing it with general trainable query attention [1], where the obtained attention scores are used to compute a weighted sum of value vectors. In this case, the trainable query (calibration) vector can be optimized towards its goal (assigning weights to value vectors) directly bacause the attention scores serve as weights directly.
> > >
> > > In our approach, the calibration vector is trained towards its goal (deriving a good sorting) in an indirect way. You can view the attention scores as unnormalized weights. When training starts, the calibration vector gives a random sorting and unmeaningful weights, so the close nodes might not be informative. Then using 1-D convolution to achieve non-local aggregation would give bad output. So, the back-propagation would influence these weights of embeddings among the same receptive fields. Note that embeddings in the same receptive field of 1-D convolution have similar weights because we sort the embeddings based on weights before convolution. Hence, if these embeddings are not from informative nodes, the corresponding weights should be influenced based on back-propagation. In this way, the calibration vector would be trained to generate good sorting indirectly.
> > >
> > >
> > > Thank you very much for providing these careful and comprehensive reviews and comments. We believe these are very helpful for our work. We hope your concerns are addressed.
> > >
> > > [1] Yang et al. "Hierarchical Attention Networks for Document Classification". NAACL HLT 2016.

---

> ### Author Response · Authors · 2020-11-14
> **Our response to Reviewer #1 (Part 1)**
>
> We sincerely thank you for the constructive comments. We answer the questions as follows.
>
> Q1: (Dis-assortative or i.i.d.): - The authors in Figure 1 - show that homophily of the created proxy graph is a value larger than that of the original graph. However, from table A.2 in the appendix - it is clear to see that MLP's outperform GNN's with or without the attention sorting in the dis-assortative graphs - and the performance of the MLP's and proposed augmented NLMLP are well within one standard deviation from each other. This questions the need to employ a proxy graph construction on top of MLP's for these datasets as it appears like the data can be treated as i.i.d (and not relational). Moreover, these datasets (used from Pei et al. 2020) are extremely small to draw any significant conclusion. Also almost no gains are seen on the assortative datasets Citeseer, Cora, Pubmed (Please add datasets from OGB) - and their running times (when augmented with a proxy graph - are the gains worthy of increased run times?).
>
> A1:
>
> (1) Yes, on several disassortative graphs (Actor, Cornell, etc), where nodes within the local neighborhood provide more noises than useful information, MLP outperforms popular GNNs that perform local aggregation. This can only show that the local aggregation hurts the performance on these graphs. We believe the nodes cannot be treated as i.i.d, and we can achieve better performance if we can aggregate information among nodes with the identical label although they might be distant with each other. This aggregation can help to distinguish the features of nodes with different labels, thus easing the following-up classification task. This point is verified by the improvement of augmenting our non-local aggregation to MLP or GNN.
>
> (2) We agree that the existing disassortative graphs are relatively small and the performance of the MLP and NLMLP are within one standard deviation on several datasets. To make it more convincing, we computed the $p$-value of the two sample t-test between MLP and NLMLP on these datasets. The obtained $p$-values, shown as the following table, are smaller than $0.05$, which demonstrates our improvement is statistically significant on these datasets.
>
> (*p*-value)
>
> Texas    Cornell  Wisconsin
>
>  0.022      0.048     0.046
>
> (3) As discussed in our paper, local aggregation is effective on assortative graphs and non-local aggregation is not necessary because it would bring more noise if distant nodes are not from the identical class. We provide the performance on assortative graphs just for reference, and we do not claim that our non-local GNNs can improve the performance on assortative graphs.
>
> Q2: (Baselines): Since the authors propose a strategy to construct a proxy graph (and the number of neighbors of each node in the proxy graph is the same??) - baselines such as creating graphs where nodes with identical labels are connected are also connected to each other / GNN on simple k-Nearest neighbors created using initial features (While a simple k-NN might appear more expensive - but the computation here is a single time effort) appears crucial. Also add a baseline, where adjacency structure of the graphs are iteratively updated during training such as - Learning discrete structures for graph neural networks (Franceschi, et al. ICML 2019)
>
> A2: Thank you for providing these interesting ideas about baselines.
>
> (1) Yes, the number of neighbors of each node in the “proxy graph” is the same, which is determined by the kernel size of the subsequent 1D convolution.
>
> (2)
> To make it more convincing, we conduct a comparison between our NLMLP with MLP+KNN+GCN, as described by reviewer #1 and #3. The results are shown in the following table.
>
> |Method               |Actor|Cornell|Texas|Wisconsin|
>
> |MLP                     |35.1  |81.6      |81.3   |84.9          |
>
> |MLP+KNN+GCN|36.8  |81.2      |81.0   |86.0          |
>
> |NLMLP                |37.9  |84.9      |85.4   |87.3          |
>
> Overall, MLP+KNN+GCN performs better (sometimes not obvious) than naïve MLP. Our NLMLP outperforms baselines consistently.
>
> Also, MLP+KNN+GCN is much more computationally expensive when the number of nodes is large since the time complexity of computing pairwise similarity is $O(n^2)$. For example, on the Actor dataset (7600 nodes), the training times per epoch (averaged over $500$ epochs) of MLP+KNN+GCN and our NLMLP are $464$ms and $8$ms, respectively.
>
> (3) Thank you for leading us to this interesting related work (Franceschi, et al. ICML 2019). This work aims to learn the graph structure and parameters in GNN jointly in the cases where graph structure is not available. Since this is not related to capturing long-range dependencies, it is not straightforward and natural to include it as a baseline. We will include this line of work into the related works in our revised version.

---

### Official Review · AnonReviewer4 · 2020-10-28
**Interesting Simple Idea, Clearly Experimental Gains**

**Rating:** 7
**Confidence:** 2

**Review:**

This paper proposes a way of speeding up non-local aggregation on graph convolutional neural networks based on sorting the nodes into an ordering, and performing a 1-D convolution on this resulting ordering. This algorithm has the advantage of being asymptotically faster than other non-local aggregation schemes, and the paper demonstrates that empirically it can do at least as well as some of the other methods.

Strengths:
+ the proposed approach is simple, quite general, and rather different from other tools for graph neural nets that I'm aware of.
+ the experimental evaluation methodology is sound, and comparisons with several previous works are made

Weaknesses:
- the approach is difficult to interpret: it's difficult to convince someone working on GCNs why it would work.
- on some of the data sets, the gains observed as inconclusive. The experiments also focused on small data sets: it's unclear how such gains extend to more general settings.

I work mostly on graph algorithms, and only know a little about neural networks. So I'm evaluating this paper mostly as a practical graph algorithms. The effectiveness of such global sorting schemes based on a single score is very surprising, almost too surprising. On the other hand, my general impression is that graph algorithms is full of such surprises: many by now classical algorithms are arrived at by analyzing strange phenomenon that happen to work well. So I'm quite willing to suspend disbelief about why something like this would work, as that's a much more detailed process.

From the discussions, it seems that there are quite a bit of concerns raised about the experimentation process. On the other hand, the responses, and presentations in the paper, are also quite convincing to me. So I believe this result is ready to appear in the conference, if anything for the further discussion/interest it will generate, and would still like to recommend acceptance of this paper.

---

> ### Author Response · Authors · 2020-11-15
> **Our response to Reviewer #4**
>
> Thank you for your positive comments.

---

### Official Review · AnonReviewer3 · 2020-10-28
**interesting and novel idea of non-local GNN for disassortative graphs**

**Rating:** 7
**Confidence:** 5

**Review:**

This paper points out an interesting and important issue of GNNs, i.e., local aggregation is harmful for some disassortative graphs. It further proposes non-local GNNs by first sorting the nodes followed by aggregation. The paper is well written and easy to follow.

+ Positives
1. The paper studies an important problem. The proposed Non-local GNNs by first sorting the nodes followed by aggregation is interesting and makes sense.
2. The paper is well written and easy to follow.
3. Experiments well support the claim of the paper. The results demonstrated the effectiveness of the proposed method for disassortative graphs for node classification. In addition, the authors show the running time to demonstrate its efficiency and analyze the sorted nodes to demonstrate that the proposed method can learn non-local graphs.

-Negative
1. It seems that for some disassortative graphs such as Actor, Cornell, Texas and Wisconsin, using the node attributes to build the non-local graph is much effective than using the attributed graph. The authors may also need to compare a baseline that simply use MLP to learn node embedding, then construct the graph by calculating pairwise node similarity, followed by GNN for node classification. This can be treated as a variants of the proposed NLMLP to show that sorting the nodes is more efficient and more effective.

---

> ### Author Response · Authors · 2020-11-15
> **Our response to Reviewer #3**
>
> Thank you for the insightful review and positive words. We address the concerns as follows.
>
> Q1: It seems that for some disassortative graphs such as Actor, Cornell, Texas and Wisconsin, using the node attributes to build the non-local graph is much effective than using the attributed graph. The authors may also need to compare a baseline that simply use MLP to learn node embedding, then construct the graph by calculating pairwise node similarity, followed by GNN for node classification. This can be treated as a variant of the proposed NLMLP to show that sorting the nodes is more efficient and more effective.
>
> A1:
> Thank you for providing this great suggestion. We conduct a comparison between our NLMLP with MLP+KNN+GCN, as described by the reviewer. The results are shown in the following table.
>
> |Method               |Actor|Cornell|Texas|Wisconsin|
>
> |MLP                     |35.1  |81.6      |81.3   |84.9          |
>
> |MLP+KNN+GCN|36.8  |81.2      |81.0   |86.0          |
>
> |NLMLP                |37.9  |84.9      |85.4   |87.3          |
>
>
> Overall, MLP+KNN+GCN performs better (sometimes not obvious) than naïve MLP. Our NLMLP outperforms baselines consistently.
>
> Also, MLP+KNN+GCN is much more computationally expensive when the number of nodes is large since the time complexity of computing pairwise similarity is $O(n^2)$. For example, on the Actor dataset (7600 nodes), the training times per epoch (averaged over $500$ epochs) of MLP+KNN+GCN and NLMLP are $464$ms and $8$ms, respectively.
>
> Thank you again for your constructive comments.

---

### Author Response · Authors · 2020-11-19
**Response to all reviewers**

Dear reviewers,

Thanks for the constructive and insightful reviews.  We provided comprehensive responses to the proposed concerns. Please kindly check it. We hope we have addressed the concerns raised by the reviewers. Thank you.

---

### Decision · Program_Chairs · 2021-01-07
**Final Decision**

**Decision:**

Reject

**Comment:**

This paper is right at the borderline: the reviewers agree it is well written, proposing a simple but interesting idea. However, there was a feeling among the reviewers (especially reviewer 1) that the paper could be strengthened considerably with a better discussion/some theory on the sufficiency of the calibration vectors, as well as experiments on larger datasets. Doing one of these would have substantially strengthened the paper. Due to the remaining shortcomings, the recommendation is not to accept the paper in its present state.